# A Low-Power Analog Cell for Implementing Spiking Neural Networks in 65 nm CMOS

**John S. Venker, Luke Vincent and Jeff Dix \***

Department of Electrical Engineering and Computer Science, University of Arkansas, Fayetteveill, AR 72701, USA; jsvenker@uark.edu (J.S.V.); lv001@uark.edu (L.V.)
**\*** Correspondence: dix@uark.edu; Tel.: +1-479-575-6052

**Abstract:** A Spiking Neural Network (SNN) is realized within a 65 nm CMOS process to demonstrate the feasibility of its constituent cells. Analog hardware neural networks have shown improved energy efficiency in edge computing for real-time-inference applications, such as speech recognition. The proposed network uses a leaky integrate and fire neuron scheme for computation, interleaved with a Spike Timing Dependent Plasticity (STDP) circuit for implementing synaptic-like weights. The low-power, asynchronous analog neurons and synapses are tailored for the VLSI environment needed to effectively make use of hardware SSN systems. To demonstrate functionality, a feed-forward Spiking Neural Network composed of two layers, the first with ten neurons and the second with six, is implemented. The neuron design operates with 2.1 pJ of power per spike and 20 pJ per synaptic operation.

**Keywords:** neural networks; neural network hardware; neuromorphic; analog computing; low-power electronics; edge computing; spiking neural networks; analog circuit design

## 1. Introduction

The search for a compact and low-power analog cell required for a hardware artificial neural network (ANN) with on-board learning continues. The appeal arises due to an ANN's ability to process large numbers (millions [1]) of signals in parallel by using simple physical laws, i.e., (Ohm's and Kirchhoff's) multiplication and summation of signals. For inference at the edge, analog neuromorphic systems promise advances in CMOS energy efficiency. Physical realizations such as Spiking Neural Networks (SNNs) outperform algorithms implemented on traditional hardware in most applications [2], particularly those in vision and pattern recognition, where only parallel processing and not training is necessary [3]. The greatest barrier to these technologies having a larger commercial impact lies with the efficiency of the analog neurons [1]. The most mature and large-scale implementations of ANNs all still use CMOS technology, despite the predication that emerging experimental devices such as memristors, photonic circuits, or ferroelectric devices could drastically improve neuromorphic systems.

There is a great deal of debate regarding the proper implementation of SSN, be they digital, analog, memory, neuromporhic, or based on new devices. This focus here is on low-precision, efficient neuromoprhic cells. This design strikes a compromise between keeping the spiking rate low enough to approximate biological signals effectively, while keeping it high enough to maintain reasonable inference accuracy. It is compatible with standard CMOS.

In this paper, the first section will be devoted to design, architectural considerations, and circuit implementations, while the second will discuss experimental methods, detail the testing procedure and criteria used, and conclude by comparing measured results with similar works.

## 2. Results

### 2.1. Training Data and Software

Non-radial training data provide a simple way to test the SNN [4,5]. These two radial, nonlinear groups of data have their X and Y values converted into respective five-bit binary, encoding what neurons on the input layer are sent a spike. These spikes are compiled into a matrix. A Python script transforms the digital input data to a temporally encoded scheme. This algorithm is implemented on a Raspberry Pi and spike data are sent to the DUT.

### 2.2. Output Spike

The first experiment was to stimulate an input neuron repeatedly until an output was observed. This was a simple heartbeat test for the chip. The energy per output spike was on average measured to be 2.1 pJ/spike. This functionality can be seen in Figure 1.

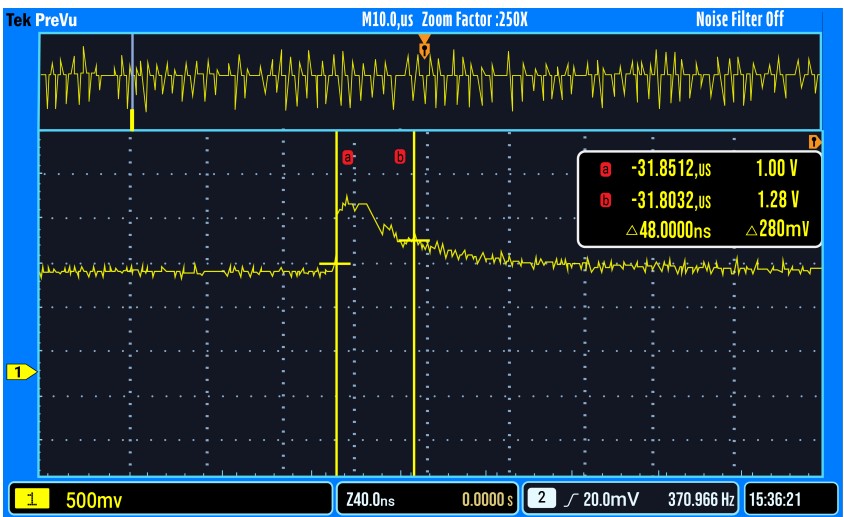

**Figure 1.** Output spike.

### 2.3. Training and Testing

The primary concept of this test (Figure 2) is to train the SNN on group one data, and then test the network on group two data (Figure 3). As the network was trained on group one data, the group two data should not cause the network to produce an output spike. The yellow waveform shows the micro-controller spikes, while the blue waveform shows the network's response. For the first four training spikes, no spike was observed, as expected, and the last spike produces an output showing that the weights have become accustomed to the trained data. For the second test, the SNN was trained using group one data, the same data it was tested on previously. The SNN should, therefore, spike often, demonstrating it is accustomed to the learned data. Accuracy is measured by sending ten training pulses from a given group, and then one random pulse from either group, with the accuracy percentage measuring how often the SSN "guessed" correctly whether to spike or not in response to the single random spike.

### 2.4. Accuracy

The main test for this work is to demonstrate just how well the SNN architecture can learn patterns in data. To do this, a certain amount of training Group 1 spikes was given to the network followed by one test data point of either Group 1 or Group 2. For each number of training spikes, the test was performed 20 times and randomly given 10 Group 1 data points and 10 Group 2 data points. If the SNN thinks the tested data point is Group 1, it will produce a spike; and it will produce no spike if it thinks it is Group 2. The following graph in Figure 4 shows the results of this experiment.

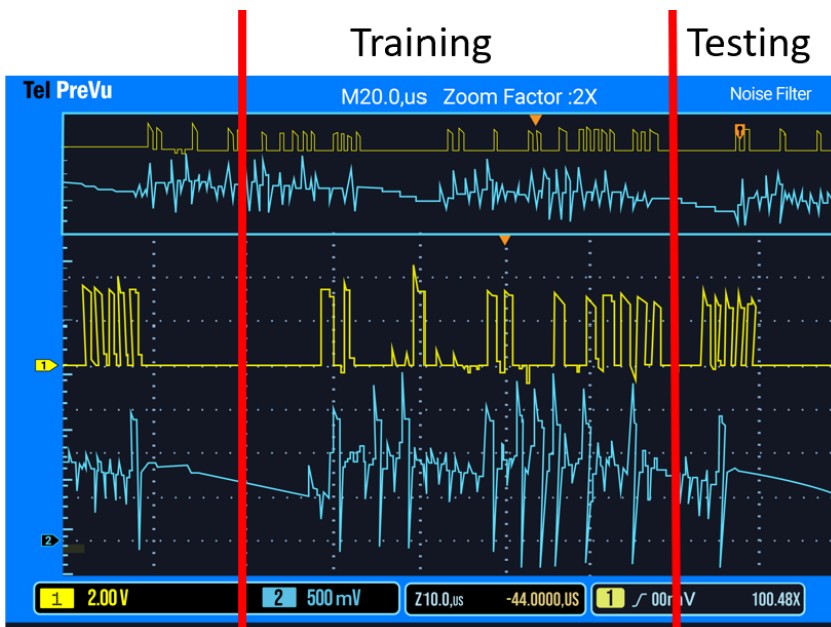

**Figure 2.** Group 1 training and testing.

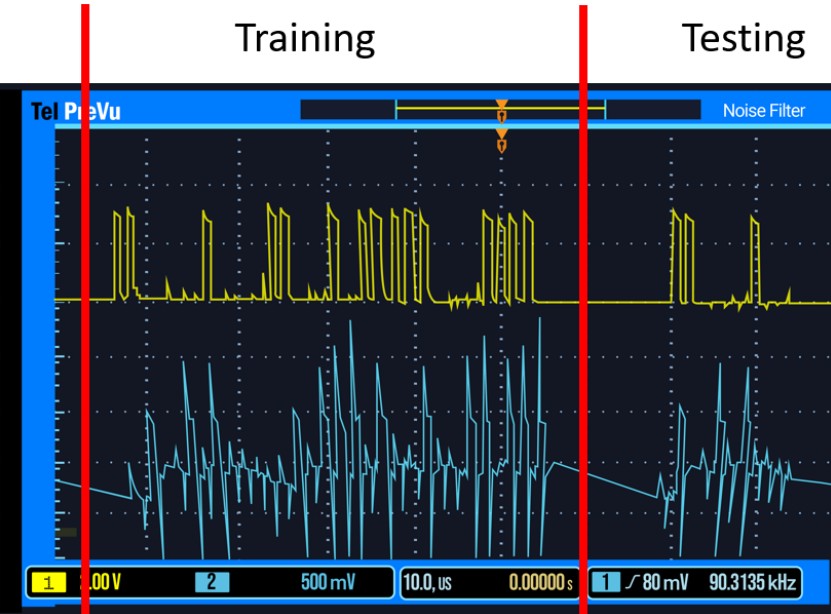

**Figure 3.** Group 2 training and testing.

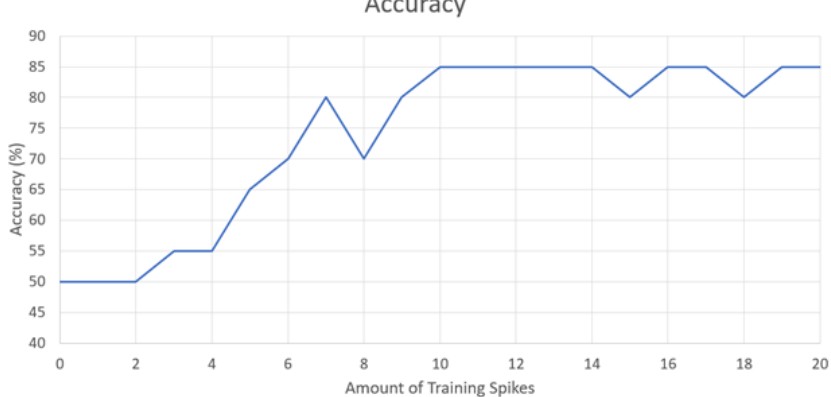

**Figure 4.** Testing data accuracy.

It is observed that the minimum accuracy with around two to four spikes achieves 50% accuracy. In this instance, the SNN has not received enough spikes to spike on the output and will always guess Group 2. As the SNN receives more training spikes (4–9), the SNN starts to learn the pattern for Group 1 data points. It begins to spike when given a Group 1 data point and will not spike when given a Group 2 data point. Diminishing returns are observed when giving the SNN 10 or more spikes. We hypothesize this to be an effect of the leakage on the synaptic weight capacitor over time in the STDP circuits. The SNN is unable to fine tune the weight parameters to achieve higher accuracy. Another hypothesis is due to the low number of neurons and synapses in the SNN architecture. More layers and neurons would offer the ability for greater inferences on the data, leading to higher accuracy. The SNN was not able to achieve higher than 85% accuracy on this test. To measure a more precise accuracy, 12 training spikes was chosen as the optimal amount of spikes to train the network, and 50 data points were used for testing the network. An accuracy of 86% was achieved through this method.

## 3. Materials and Methods

A shallow, strictly feed-forward SNN of an input layer, a single hidden layer, and an output layer is used to demonstrate the functionality and efficiency of the developed SNN cell. The number of neurons per layer is shown below in Figure 5, where each blue line represents an individual synapse to record the activity between neurons. The neurons themselves are represented by circular symbols with a spike in the middle.

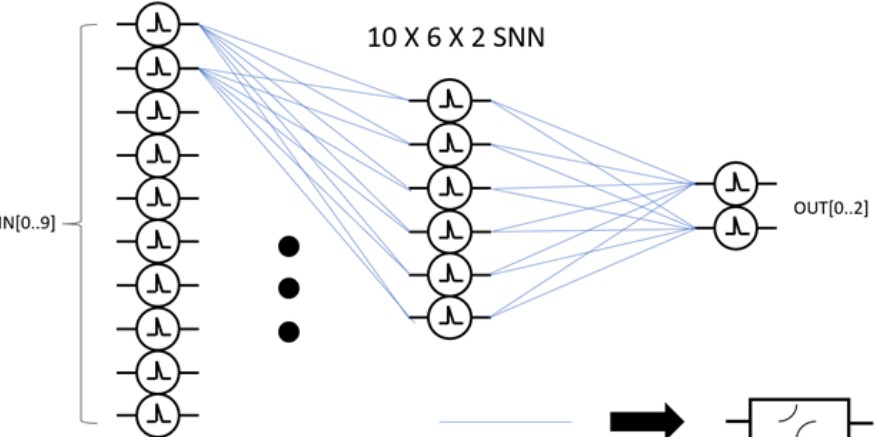

**Figure 5.** SNN architecture.

As this was a multi-project chip, the primary constraint for the size of the network was the number of pins available on the chip. The primary architectural decisions were to use a spike-timing-dependent architecture. This implementation of an SNN encodes information in the precise timing of a spike, where neurons spiking in close proximity to another allows for the value of the synapse weights to change, otherwise remaining the same. The 65 nm node was chosen due to its compromise between fast transistors, its area, and its ability to implement analog circuits. This node does not contain the multi-patterning issues of smaller nodes, their short-channel limitations, or only having discrete drive strengths as in FinFet technologies [6].

### 3.1. Leaky Integrate and Fire Neuron

The leaky integrate and fire neuron imitates the functions of the soma, dendrites, and axons of a biological neuron [7]. Input spikes are accumulated on the captive membrane node, Vmem in Figure 6. After the charge rises above the threshold set by Vref (250 mV), the LIF provides an output signal via a comparator designed to provide a biologically similar spike [8] with a hysteresis of 50 mV. After firing, the membrane is depleted to its resting potential of around 100 mV.

As seen in Figure 7, a 1 ns spike width is chosen for illustrative purposes. As successive spikes are integrated, and the membrane charge builds, the LIF neuron fires a spark after the threshold is reached, and the membrane potential resets. A metal-oxide-metal capacitor is used to implement C1, the membrane. The lengths and widths of all PMOS and NMOS transistors used in this circuit are kept to minimum aspect ratios of widths equal to 200 nm and lengths equal to 60 nm.

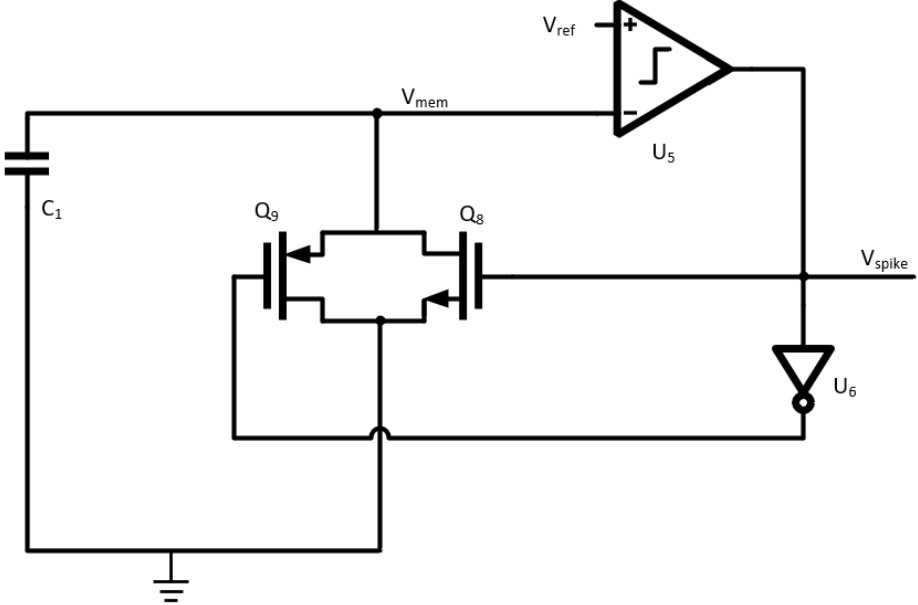

**Figure 6.** LIF circuit.

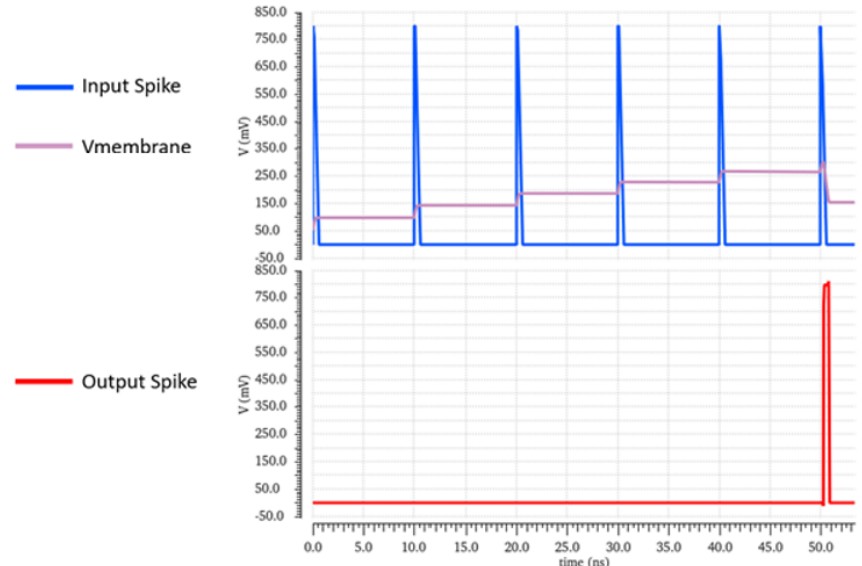

**Figure 7.** LIF input spike (blue), weight (grey), and output spike (red).

### 3.2. Comparator Circuit

The comparator (Figure 8) is a key block for the LIF neuron. Its design is kept simple to achieve a high-density, low-power layout [9]. The comparator uses four inverters to delay the signal, allowing the milli-Volt spike membrane voltage to drain. This mechanism establishes the mandatory refractory period between spikes. The comparator core is an operational transconductance amplifier consisting of a differential pair that compares the value of a reference on one leg of the differential pair with the input value of another. The current from the left leg is mirrored to the right in order to form the output. The output

is further mirrored by a second push-pull stage which can sink and source more current to drive a capacitive load, such as an inverter's input and increase swing to improve speed and decrease the effect of noise.

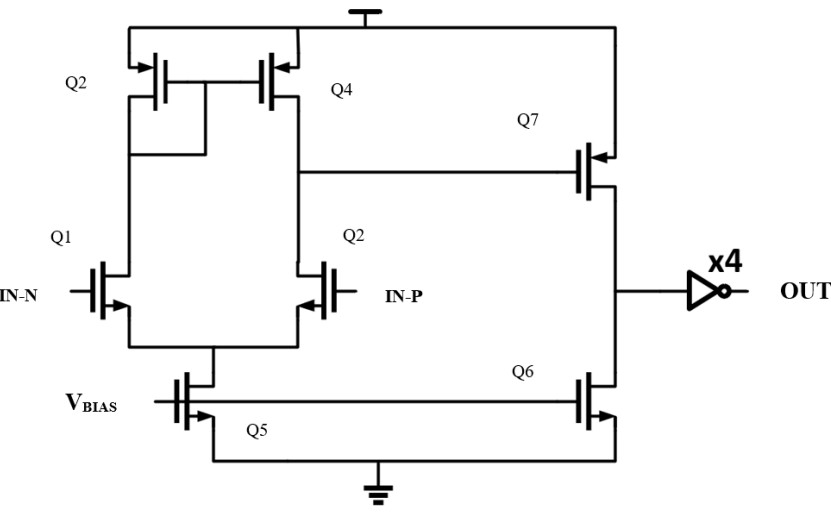

**Figure 8.** Comparator circuit.

### 3.3. Spike Timing-Dependent Plasticity

Spike timing-dependent plasticity allows for large-scale pattern recognition [2,10]. As seen in Figure 9, by changing the membrane weights when a pre-synapse signal arrives before a post-synapse (which leads to a decrease in the likelihood of that path being chosen in the future or vice versa for the converse scenario), intelligent decisions can be arrived upon. The strength of the change is a function of the time difference between the spikes:

$$W(\Delta t = t_{pre} - t_{post}) = \begin{cases} A_+ \exp \frac{\Delta t}{\tau_+}, & \text{if } \Delta t \geq 0 \\ A_- \exp -\frac{\Delta t}{\tau_-}, & \text{if } \Delta t < 0 \end{cases} \tag{1}$$

where $W$ is the learning function, $\tau_-$ and $\tau_+$ are the range in time of the pre-synaptic and post-synaptic interval, and $A_+$ and $A_-$ represent the maximum synaptic modification for either depression or potentiation, both of which occur near $\Delta t = 0$.

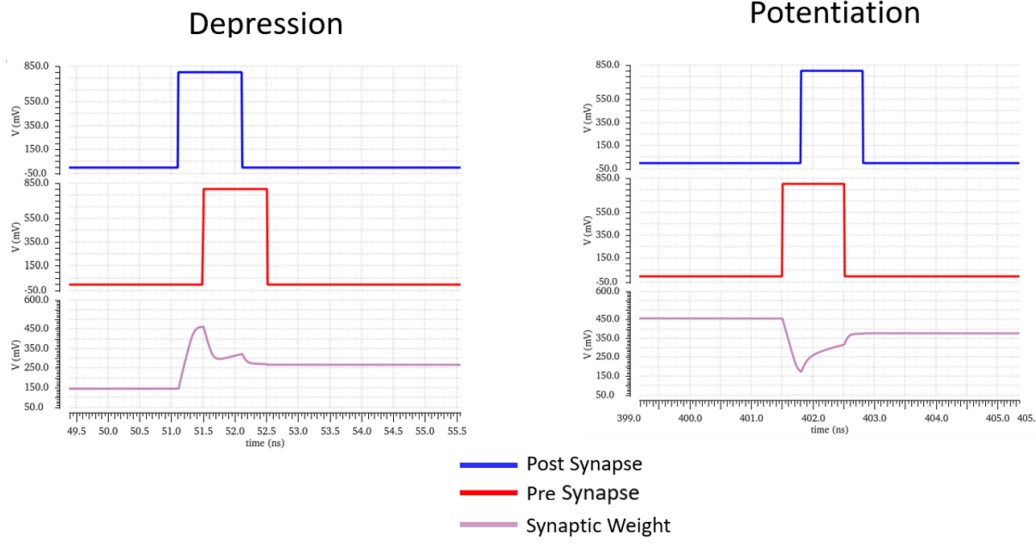

**Figure 9.** Depression and potentiation.

The circuit proposed for the synaptic connection must only change its weight value, held on capacitor C4, when both neurons before and after it in the SNN architecture fire. In Figure 10, the top (mirrored on the bottom) structure formed by Q10, Q15, Q16, and C2 is an integrator holding the spike's value. One integrator holds the value of the pre- or post-synaptic spike, simultaneously activating a buffer that disables the other integrator, so it does not disturb the circuit functionality. Additional transistors are added to assist in the depression of the weight capacitor value, increasing the speed at which the SNN learns by responding more quickly to uncorrelated spikes [7].

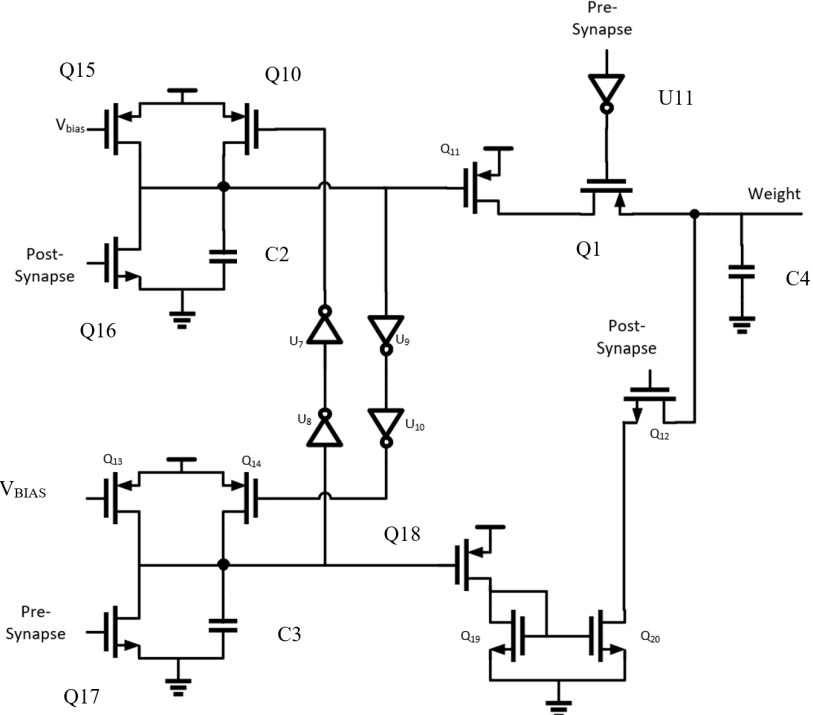

**Figure 10.** Spike timing-dependent plasticity circuit.

In the situation where the pre-synaptic spike is received first and the post-synaptic spike later, then the weight should decrease on the weight capacitor. Q17 is active, depleting C3 and putting Q18 into triode. This biases the current mirror, decreasing the weight value. The procedure for increasing the weight value is similar, with a path from VDD being formed via Q1 and inverter U11 activating when the pre-synaptic pulse occurs.

### 3.4. Activation Function

A sigmoid or logistics curve is a common activation function used for machine learning applications that a PMOS transistor's transfer function resembles. This non-linear activation function can be realized in a variety of ways, the most efficient of which was found to be a single PMOS with the weight function applied to the gate. The STDP circuit operates with a voltage range of 0 V to 600 mV, and the designed PMOS covers this range to 100–800 mV allowing for the neuron's recorded values to be kept in a range well-suited for the architecture. Figure 11 details this functionality.

### 3.5. SNN Architecture and Layout

The spiking neural network architecture chosen for the design was a 10 by 6 by 2 composition. This composition means that there will be 10 inputs that are capable of receiving and transmitting asynchronous analog spikes. These inputs will be sent to the first layer of 10 neurons that will accumulate charge and fire signals throughout the rest of the SNN architecture. The hidden layer will be composed of 6 neurons. These will form the non-linear inferences from abstract data. There is an STDP circuit represented by each blue

line in Figure 5. Because there is an STDP circuit for every possible connection between neurons in adjacent layers, the area for SNNs increases exponentially with the number of neurons in the architecture.

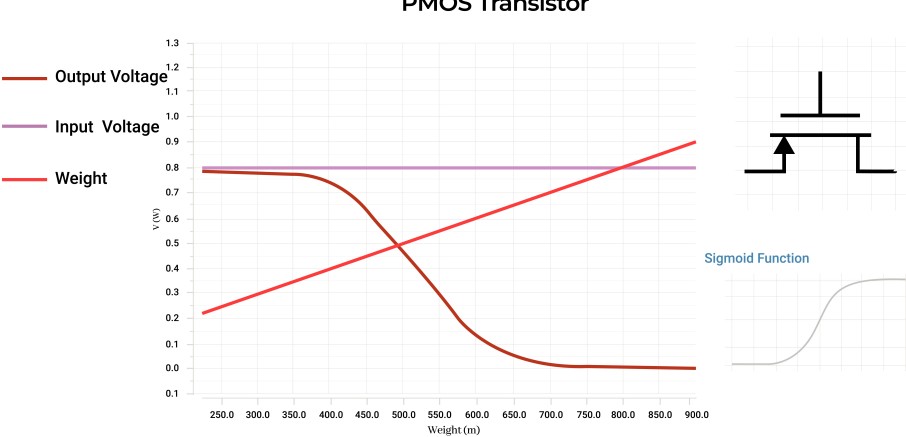

**Figure 11.** Activation function circuit and operation.

The second layer, also known as the hidden layer, will accumulate charge and send signals to the inputs of the final layer of two neurons. These neurons will then send their outputs off-chip signifying the answer to the proposed problem of 10 input spikes. The first-layer neuron number was determined by the integrated circuit input/output pin limitations with the feed-forward SNN created around this limitation. The number of neurons in the hidden layer was determined through simulation optimization of the proposed SNN through the Brian2 python library. This library is an SNN simulator that offers the physical calculations of electrical properties in an SNN. Six neurons in the hidden layer yielded the best performance during simulations. The output layer was also limited by the amount of input/output pins on the integrated circuit. Two neurons were provided on the output to offer the capability of four different SNN outcomes.

Figure 12 details the layout with 10 neurons on the first layer in the blue box followed by 60 STDP nodes that connect each neuron on the first layer to each neuron in the second layer. The red box shows the hidden layer containing six neurons. Then, there are 12 STDP nodes that connect each neuron on the hidden layer to each neuron on the output layer. The layout was constructed to be as compact as possible to demonstrate the high-density capabilities of the SNN and is only 253.82 µm by 155.2 µm.

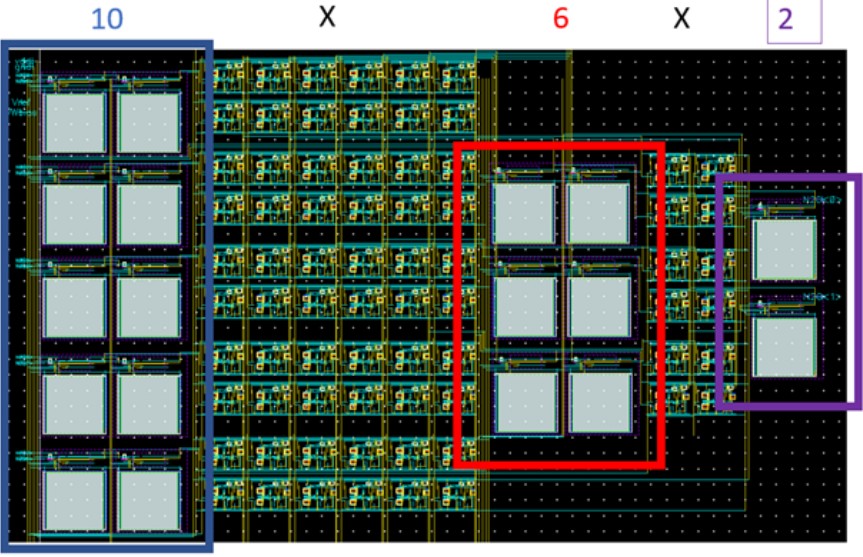

**Figure 12.** SNN layout zones.

## 4. Discussion

The SNN architecture is built within a commercially available 65 nm standard Silicon CMOS-integrated circuit technology node. The 65 nm process node is chosen to offer the greatest flexibility in terms of analog/mixed-signal circuit design within a mature process node while attempting to push an analog design to more digital-friendly design processes. The 65 nm process does benefit a subthreshold design with larger threshold voltage devices, leading to improved power efficiency at the cost of speed. The consequence of the more mature process node is that low-power, high-speed digital devices for some of the STDP and neuron design are not as available as would be in newer, more digital technology nodes.

The most relevant metric to the VLSI cell created is the energy consumption from a single spike event [11]. This metric directly relates to if the SNN could perform well despite the large number of synaptic connections needed to realize networks capable of solving complex problems.

Table 1 details the neuron energy consumption comparison with literature. The total power is measured by applying pulses to 25% of input neurons in line with the methodology of [12] and measuring quiescent power at the output. By using a comparator instead of a transconductance amplifier, the cell we realized has a faster recovery and spike rate in exchange for additional power usage.

**Table 1.** Neuron energy consumption comparison.

| Work | Power | Rate [Spike/s] | Energy per Spike (pF) |
|---|---|---|---|
| Zhang [13] | - | - | 480 |
| Tanaka [14] | 125 µW | 3M | 83 |
| Cruz-Albrecht [7] | 37 pW | 100 | 0.37 |
| This Work | 4.5 µW | 2.5M | 2.1 |

The most important metric in Table 2 is the power-per-synaptic-operation, as it measures not only the power of a single spike, but also the power the STDP circuit consumes during operation of a single node of the network [12]. Without individual testing structures and input/output pins for the fabricated chip for the STDP and neuron by themselves, the power metric was calculated by measuring the current drawn by the circuit in the test setup alongside the voltage levels seen from the waveforms. This measurement was divided by the number of operations during the testing period to obtain the power per operation statistic, seen in Table 2. From this result, it can be seen that as the number of synaptic connections increased, reasonable accuracy could feasibly be achieved using the proposed cell. To realize such a large number of neurons, the addressed event representations are needed to communicate between the large numbers of neurons [1,3,15]. To implement such a system, memory, a communication protocol, and digital-to-analog interfaces would need to be constructed, which is outside the scope of this initial work.

**Table 2.** VLSI SNN comparisons.

| Work | Technology | Neurons | Synapses | Power/Operation | Accuracy |
|---|---|---|---|---|---|
| BrainScaleS [4] | 130 nm | 180 k | 40 M | - | 95% |
| Odin [11] | 28 nm | 256 | 64 k | 12.7 pJ | 84.5% |
| Loihi [10] | 14 nm | 130 k | 130 M | 23.6 pJ | 99.91% |
| This Work | 65 nm | 18 | 72 | 20 pJ | 86% |

The SNN architecture for this work proved to be successful in learning complex non-linear data patterns and could, therefore, be implemented in many deep learning neural network applications; however, the accuracy for the architecture did not compare to the

work of other SNNs developed in the literature. This is hypothesized to be attributed to the lack in number of neurons and synapses. It is also attributed to the short-term memory of the weight values in the synaptic weight of the STDP circuits. Other papers [16,17] have used SRAM for storing the weight values digitally to prevent the decay of the synaptic weight. Making the aforementioned changes would theoretically make this design more accurate and more plausible for integration with modern-day technologies such as smart phones. The energy consumption on the device was accurate to the simulations, and if it were scaled up with the changes mentioned, there is potential for very low power applications of deep neural network methods.

### 5. Conclusions

An efficient, compact analog VLSI cell is first developed and tested, then compared with relevant work. New circuit topologies are employed to increase performance without significantly sacrificing power consumption. Comparison with contemporary work shows these criteria are met, and a large-scale system could be realized with the analog cell developed. The SNN architecture presented has the capability to be expanded significantly to provide an effective computing architecture within many edge devices that have stringent power limitations. The presented cell offers relatively accurate processing capabilities with good power per operation seen all within a cheap and mature, 65 nm commercially available technology node. Additional constructs would be required to provide data to and from the integrated circut (analog-to-digital or digital-to-analog converters), but these constructs can readily be designed and optimized within the chosen technology node. Furthermore, the design's input/output pin limitations as well as area constraints prevented it from achieving greater accuracy and power efficiency performance metrics. Expanding the SNN architecture to include memory for weight storage and more neurons/synapses could potentially lead to a significantly improved system. Lastly, reproducible results were not possible, as one of the researchers graduated during this research and did not publish their Python testing library for open-source use within the research group. This consequence lead to protracted efforts in figure readability and data gathering.

**Author Contributions:** Conceptualization, L.V. and J.D.; methodology, L.V.; software, L.V.; validation, J.S.V., L.V. and J.D.; formal analysis, L.V.; investigation, L.V.; resources, J.S.V., L.V. and J.D.; data curation, L.V. and J.D.; writing—original draft preparation, L.V.; writing—review and editing, J.S.V. and J.D.; visualization, L.V.; supervision, J.D.; project administration, J.D.; funding acquisition, J.D. All authors have read and agreed to the published version of the manuscript.

**Funding:** This research received no external funding.

**Institutional Review Board Statement:** Not applicable.

**Informed Consent Statement:** Not applicable.

**Data Availability Statement:** No new data were created or analyzed in this study. Data sharing is not applicable to this article.

**Acknowledgments:** The authors would like to thank the University of Tennessee at Knoxville for their support and facilities in testing this work, the Integrated Circuits and Systems Lab at the University of Tennessee, and colleagues for their continual feedback and support during the design and testing of this work.

**Conflicts of Interest:** The authors declare no conflict of interest.

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
