# Peer review of "A Low-Power Analog Cell for Implementing Spiking Neural Networks in 65 nm CMOS"

_jlpea, doi:10.3390/jlpea13040055_

Round 1

Reviewer 1 Report

After reading the introduction, it seems that the goal of this work is to implement a spiking neural network into artificial neural networks. However, you should explicitly point out the significance/advantages of this work compared to other works. What are the challenges in this area of study? Point out the selling point.

Did not elaborate how the power, energy, or accuracy (listed table 1 and 2) was measured.

The presentation of data is unprofessional. Graphs are clearly took by screenshot from word editing softwares. Some contents in the figures are illegible. Photos of the instrument screen were directly used. Tables are not aligned.

Reviewer 2 Report

This is indeed an intriguing piece of research. The study of SNN is an important and attractive topic. The utilization of more advanced fabrication processes would undoubtedly enhance the significance of the study. Additionally, addressing power consumption is a commendable aspect. The work showcases clear advancements over relevant prior research.

However, the manuscript does have some major issues that need to be addressed:

1.            The author has not provided a description of how the VLSI cell with 65 nm CMOS was fabricated.

2.            There is a lack of detail on how the VLSI cell was connected, set up, and programmed to perform the tests on the oscilloscope.

3.            The author has not provided a clear description of how power consumption and accuracy were measured.

Furthermore, there are some minor issues to be addressed:

4.            The image quality appears to be suboptimal. For instance, Figure 2 seems to be a screenshot with warnings from spell correction. Additionally, the numbers in Figures 3, 5, and 7 are quite blurry. It would be advisable to recreate these figures with higher resolution.

5.            Figures 8-10 could benefit from outputting the data, replotting properly, and ensuring that the oscilloscope's output function, if available (as it should be), is used. This would significantly enhance clarity and professionalism.

6.            Abbreviations such as VLSI, CMOS, PMOS, and NMOS have not been defined in the manuscript and should be clarified for the reader's understanding.

Round 2

Reviewer 1 Report

It is concerning that the results cannot be replicated and the authors cannot not elaborate how the power was measured.

In addition, as pointed out in the previous report, some contents in the figures are still illegible. It is unprofessional to use the photos of the instrument screen.

Reviewer 2 Report

Thank you for your response.

1, While I acknowledge that the CMOS circuits process is well-understood, the manuscript asserts a connection between power and efficiency. To thoroughly evaluate this claim, it would be beneficial to include details about the specific manufacturers used and the die layout. Understanding these aspects is crucial for assessing potential impacts on power consumption and efficiency. The paper should delve into the contribution of the process versus the design in achieving the claimed power consumption benefits, which is currently lacking in the discussion.

2, If authors can include a microscope photo depicting the actual VLSI cell/chip on the PCB board, it would enhance the visual understanding of the setup.

3, Unfortunately, the authors did not incorporate changes based on my earlier points 1, 3, and 5.

4, Furthermore, the data presentation in the experimental section is not presented professionally.  making it challenging for any reader interested in replicating the work.

In summary, My suggestion is a major revision that adheres closely to the claim, supports it with well-organized and professionally presented data, and discusses the scientific and engineering perspectives underlying the claimed improvements.

Round 3

Reviewer 1 Report

The added session describing the SNN architecture and layout helps readers to understand the setup.

1. While there is much improvement on the data presentation, figure 7 is still not legible, especially the Sigmoid Function.

2. Figure 9 is still a photograph of the screen of an oscilloscope.

Reviewer 2 Report

OK this looks better now.  

The design part is good. 

Hope the follow up research can be more sophisticated on the testing part and scientific discussion.
